# Finite Element Modeling of Dynamic Properties of the Delta Robot with Base Frame

**DOI:** 10.3390/ma15196797

**Published:** 2022-09-30

**Authors:** Jakub Grabiec, Mirosław Pajor, Paweł Dunaj

**Affiliations:** 1Faculty of Mechanical Engineering and Mechatronics, West Pomeranian University of Technology in Szczecin, Al. Piastów 19, 70-310 Szczecin, Poland; 2FANUC Polska Sp. z o.o., Tadeusza Wendy 2, 52-407 Wrocław, Poland

**Keywords:** delta robot, finite element analysis, dynamic properties, Guyan reduction

## Abstract

The paper presents the finite element modeling of the dynamic properties of a delta robot attached to a steel frame. A distinguishing feature of the proposed modeling method is the application of the Guyan reduction method in modeling of frame foundations. The frame in question was analyzed in two variants: (i) without attached robot and (ii) with attached robot. Based on the established model, the dynamic properties (i.e., natural frequencies, mode shapes, and frequency response functions) of the frame in the two variants were analyzed. The obtained results were then experimentally verified and validated. It was found that the developed model showed an average relative error for natural frequencies of 4.3% in the case of the frame and 5.6% in the case of the frame with the robot. The paper demonstrated the validity of the proposed model, allowing accurate and fast determination of robotic system dynamic properties.

## 1. Introduction

Parallel delta robots have been widely used in various industries, especially in pick-and-place applications [1], which rely on the cyclical movement between defined positions [2]. To achieve high efficiency, the cycle times are minimized, which in turn causes the appearance of substantial dynamic loads [3]. Those time-varying loads frequently cause vibrations that adversely affect the accuracy and repeatability of the robot’s positioning [4]. The minimization of those vibrations is most often performed by a proper robot design [5] or by the optimization of the robot’s trajectory [6]. However, a reliable method for modeling dynamic properties is needed to properly design a robot structure or to perform its trajectory optimization [7].

Kermanian et al. [8] developed formulations for the dynamic analysis of parallel robot considering its links flexibility. These formulations were based on: (i) the co-rotational finite element method (CRFEM) and (ii) rigid finite element method (RFEM). The first method involved using CRFEM to describe the deformation of each element directly in its co-rotated frame without the need of expressing any intricate kinematic relations. The second one, based on the RFEM, made it possible to simplify the form and derivation of kinetic energy of a flexible link. The proposed methods were used to study the dynamics of a circular and non-circular pick and place trajectory. The comparison of the obtained results showed that between the two methods for the same number of nodes the results differed in the worst case by 6%. However, in terms of computational costs, the RFEM based method always reduces the simulation time by 31–46%. It was therefore concluded that presented methods allow for efficient analysis of dynamic properties of a parallel robot with the geometrical nonlinearities that occur to flexible links due to the large deformation.

Zhang et al. [9] presented the finite element analysis of the dynamic performance of a modular reconfigurable parallel robot. To assess the dynamic properties of the robot, the modal analysis was conducted, and the frequency response functions were determined for four different configurations. Based on the resonance frequencies, amplitudes, and the main vibration direction of each configuration the weak links of the robot system were found and the robot performance under the action of external forces was assessed. In conclusion, a theoretical basis for the design of the robot system with higher precision was developed.

Righettini et al. [10] proposed a new method for determining the mode shapes of the linear delta robot [11]. The presented approach consisted in building a model that combines the properties of a delta robot and a belt transmission of a prismatic joint. The established model consisted of a rigid part representing the inertia of the delta robot and a configuration-dependent lumped stiffness element representing the belt transmissions. The model-based modal analysis conducted showed that the system natural frequencies and the directions of the end-effector modal displacements strongly depend on its position in the working space. The proposed approach constitutes a useful support for the system design in evaluating the end-effector vibration direction for a given vibration mode.

Shehata et al. [12] presented a procedure for parameters estimation in multibody system model of the delta robot. The model was based on Lagrange formulation and elaborated using Matlab Simscape Toolbox. The identification module was applied to estimate parameters of the model by comparing the simulated model output with experimental motor measurements. The developed model was used to predict end-effector displacement, angular velocities of the motors and the reaction forces and torques acting on the base frame due to joint rotation. It was concluded that the developed procedure can be used to identify and optimize the structure of the delta robot mechanism.

Yang et al. [13] proposed the modification of a delta structure to improve the loading stiffness of the robot over the entire workspace. The modification consisted in applying a non-spherical ball-socket joint. An analytical model that included a link flexibility and a joint clearance was derived. Then, the loading deflection as a function of these parameters was calculated. The developed model provided the necessary data to assess the non-spherical ball-socket joints’ influence on the total variation of the loading deflection. The experimental results show that the non-spherical ball-socket joint design reduces the total variation in the loading deflection of the end effector by up to 81% compared to that of a traditional robot with spherical ball–socket joints when performing positioning over the entire workspace at the assigned height. The proposed modeling methodology is to be used in the design leading to the improvement of positioning accuracy.

The latest research deals with a very wide spectrum of problems related to the dynamics of parallel kinematic robots. However, the literature barely mentions the experimental verification of natural frequencies, mode shapes, and frequency response functions to describe its dynamic properties. Moreover, little work is devoted to the analysis of the operation of the entire base-robot-object system. An industrial robot placed on a specially designed base, together with the load to be carried, forms a BRO system with specific dynamic characteristics. The design of the base is usually left to the end user of the robot themselves. The dynamic properties of the base largely determine the dynamics of the entire BRO system which was experimentally confirmed by Cheng and Li [14].

The aim of this paper is to develop an effective method of modeling the dynamic properties of a robotic system, especially concerning the dynamic properties of a frame. The developed methodology may serve as a tool for designing robot frames focused on minimizing the robot’s vibrations by changing the frame structure. The main novelty of the presented paper is the use of Guyan reduction for modeling the foundation of the robot frame. The distinguishing feature of the article is also carrying out a full two-stage (for the frame itself and for the frame with the robot) verification of the dynamic properties of the structure.

The paper is structured as follows: Section 2 presents the research object, a delta robot attached to a steel frame. This section also contains the finite element model building procedure and experimental test stand. In Section 3, the obtained results based on a model and experiment are compared. This Section also contains an appropriate discussion. A summary and key conclusions are provided in Section 4.

## 2. Materials and Methods

### 2.1. Analyzed Object

The analyzed object is a FANUC M-3iA/6A delta robot (FANUC Corporation, Yamanashi, Japan) intended for picking, packing, and assembly operations attached to a steel frame. The mass of a robot was 175 kg.

The delta robot under analysis is a five degrees of freedom space mechanism with three rotating legs and an end-effector with two swivel axes. Each leg is connected to the fixed platform and the movable platform by means of a set of spherical joints. The motion of the robot is controlled by three motors mounted on the fixed base that actuate the three legs to realize the movement of the moving platform.

The frame to which the robot is attached is made of the steel profiles: (i) vertical—square hollow section with dimensions 150 × 150 × 8 mm, (ii) horizontal—square hollow section with dimensions 120 × 120 × 8 mm, (iii) braces—steel tubes of diameter 108 mm and wall thickness 5 mm and (iv) ring supports—square hollow section with dimensions 100 × 100 × 5 mm. The steel ring to which the robot is mounted is made of 30 mm thick sheet metal. The frame itself is attached to the floor with the use of special anchors. Such connection ensures that the position of the frame remains unchanged during the operation of the robot. The height of the frame is determined by the range of the robot and the fact that the robot is ultimately designed to perform pick-and-place operations between two belt conveyors placed under it. The frame was designed based on the criterion that the deflection of the frame with the attached robot should be less than 0.5 mm. As a result, a generalized stiffness was obtained, determined from [15] in the X and Y directions at $7.14\times 10^{3}\;N/\mathrm{mm}$ and $17.2\times 10^{3}\;N/\mathrm{mm}$ in the Z direction. The weight of the frame was 828 kg. The structure of a steel frame with the delta robot attached is depicted in Figure 1.

### 2.2. Finite Element Model

The finite element models of the objects in question were built using the Midas NFX 2018 R1 preprocessor [16] (Midas Information Technology Co., Ltd., Seongnam, Korea). The discretization of the frame and main body of the delta robot was carried out according to the methodology presented in [17]. A structured mesh built from eight-node, cubic, isoparametric finite CHEXA elements and six-node, five-sided, isoparametric CPENTA elements was used. The utilized finite elements were characterized by linear shape functions and three translational degrees of freedom in each node.

The delta robot legs were discretized using the one-dimensional, two-nodded beam elements based on the classical beam theory (CBAR). The beam elements were characterized by a linear shape function and six degrees of freedom in each node (three translational and three rotational).

The frame foundation was modelled by using the structured mesh composed of the CHEXA elements. Since the foundation is built with different materials, the mesh sets representing different material domains were connected using the nodes coincidence.

The linear–elastic isotropic material model (MAT1) was used in finite element formulation. The material properties used were presented in Table 1, (nominal values were used in finite element modeling), and the detailed procedure for their determination can be found in [18].

To describe the damping properties of the modelled frame a complex stiffness model was used, according to which the damping matrix C can be expressed as [19]:(1)C=iηK,
where: K—model stiffness matrix; i—imaginary unit, η—loss factor.

To sum up, the developed frame model consisted of 80,274 degrees of freedom and 21,886 finite elements and for the frame with robot 164,166 degrees of freedom (80,274 degrees of freedom for the frame and 83,892 for the robot) and 64,334 finite elements (21,886 finite elements for the frame and 22,156 for the robot). The discrete models for the frame and the frame with the robot are shown in Figure 2.

The frame foundation model was reduced to a set of springs, to which the frame was then attached. The springs preserved the exact stiffness of a foundation model. The purpose of applied model reduction was to find the m degrees of freedom (DOFs) system preserving the dynamic characteristics of the full model with n DOFs, in which m<<n. The commonly used approach is to approximate the state vector q by means of transformation q=TqR, where the transformation matrix Tϵℝn×m and degrees of freedom vector qRϵℝm×1:(2)MRq¨R+KRqR=fR,
where the reduced mass MR and stiffness KR matrices are:(3)MR=TTMT,
(4)KR=TTKT.
where: K—full model stiffness matrix; M—full model mass matrix.

To select the suitable reduction method a sensitivity analysis was performed [20]. It included the determination of the significance of the impact of changing the parameters describing the foundation model on the dynamic properties of the considered structures. Consequently, it was found that the significant parameters are the moduli of elasticity and the loss coefficients for concrete and resin. Thus, the Poisson’s ratios and the densities have a negligible effect on the dynamic properties of the analyzed frame. With the above in mind, it was stated that the use of Guyan reduction is a reasonable solution due to the precise reduction in the stiffness of the structure [21,22].

The applied Guyan reduction [23] consists in reducing the degrees of freedom of the finite element model to degrees of freedom located on the border of the model (master DOFs). The reduction was made by removing slave degrees (indicated with subscript s) of freedom defined as being outside the model boundary while master DOFs (indicated with subscript m) retain the stiffness and inertia of the model. Partitioning of the state vector in the master and slave DOFs allows divided system matrices into submatrices as follows:(5)$$\left[\begin{array}{cc} M_{mm} & M_{ms}\\ M_{sm} & M_{ss} \end{array}\right]\left[\begin{array}{c} {\ddot{q}}_{m}\\ {\ddot{q}}_{s} \end{array}\right]+\left[\begin{array}{cc} K_{mm} & K_{ms}\\ K_{sm} & K_{ss} \end{array}\right]\left[\begin{array}{c} q_{m}\\ q_{s} \end{array}\right]=\left[\begin{array}{c} f_{m}\\ f_{s} \end{array}\right],$$
where: ${\ddot{q}}_{m},q_{m}$—master degrees of freedom—acceleration, and displacement respectively; ${\ddot{q}}_{s},q_{s}$—slave degrees of freedom—acceleration, and displacement respectively; $K_{mm},\;K_{ms},K_{sm},\;K_{ss}$—sub-matrices representing the associated degrees of freedom stiffness; $M_{mm},\;M_{ms},M_{sm},\;M_{ss}$—sub-matrices representing the associated degrees of freedom stiffness.

Solving the second row in Equation (5) for $q_{s}$ results in:(6)$$q_{s}=-K_{ss}^{-1}[M_{sm}{\ddot{q}}_{m}+M_{ss}{\ddot{q}}_{s}+K_{sm}q_{m}],$$
assuming that there are no loads acting on slave degrees of freedom ($f_{s}=0$) and neglecting the inertia terms results in the transformation of the state vector for Guyan reduction:(7)$$\left[\begin{array}{c} q_{m}\\ q_{s} \end{array}\right]=\left[\begin{array}{c} I\\ -K_{ss}^{-1}K_{sm} \end{array}\right]q_{m}=T_{G}q_{m},$$
where: $T_{G}$—the Guyan transformation matrix; I—identity matrix.

In the analyzed case, the reduction of the foundation model resulted in a decreasing in the number of degrees of freedom from 6468 to 120 (for a single foundation model). This is a 98% reduction of the foundation model dimensionality. Moreover, it should be noted that the mean difference between the full model and the reduced model in both analyzed variants was less than 1%. Therefore, it can be concluded that the applied reduction has a negligible impact on the accuracy of the model.

Next, a dynamic properties analysis was performed, i.e., the natural frequencies and mode shapes were determined as well as the frequency response functions. The determination of modal model (natural frequencies and mode shapes) was carried out by solving the eigenproblem formulated as follows:(8)$$\left(K-f_{n}{}^{2}M\right)\Phi_{n}=0$$
where: M—mass matrix of the model; K—stiffness matrix of the model; $f_{n}—n$-th value of natural frequency,$\;\Phi_{n}$—n-th natural vector. The calculations were conducted using Nastran Solver (SOL103).

The frequency response functions were determined using a direct frequency response method. In general, the procedure starts with the equations of motion and assumes an oscillating load:(9)$$M\ddot{q}\left(t\right)+C\dot{q}\left(t\right)+Kq\left(t\right)=f\left(\omega \right)e^{i\omega t},$$
where: ω—exciting frequency, $\ddot{q},\dot{q},q$—vectors of degrees of freedom—acceleration, velocity, and displacement respectively, f—external force vector.

Then proposing the solution q(t)  is also in the form of an oscillating function:(10)$$q\left(t\right)=y\left(\omega \right)e^{i\omega t},$$
where: y(ω)—complex displacement vector. The velocity $\dot{q}\left(t\right)$ and acceleration $\ddot{q}\left(t\right)$ is determined by taking the derivative:(11)$$\dot{q}\left(t\right)=i\omega y\left(\omega \right)e^{i\omega t}$$
(12)$$\ddot{q}\left(t\right)=-\omega^{2}y\left(\omega \right)e^{i\omega t}.$$

Substituting Equations (10) and (12) into Equation (9):(13)$$-\omega^{2}My\left(\omega \right)e^{i\omega t}+i\omega Cy\left(\omega \right)e^{i\omega t}+Ky\left(\omega \right)e^{i\omega t}=f\left(\omega \right)e^{i\omega t},$$
and dividing Equation (13) by $e^{i\omega t}$, the following equilibrium is obtained:(14)$$(-\omega^{2}M+i\omega C+K)\;y\left(\omega \right)=f\left(\omega \right).$$

On this basis, receptance y(ω), mobility $\dot{y}\left(\omega \right)$, and accelerance $\ddot{y}\left(\omega \right).$ functions are determined:(15)$$y\left(\omega \right)=f\left(\omega \right){\left(-\omega^{2}M+i\omega C+K\right)}^{-1}\;,$$
(16)$$\dot{y}\left(\omega \right)=\omega \;f\left(\omega \right){\left(-\omega^{2}M+i\omega C+K\right)}^{-1},\;$$
(17)$$\ddot{y}\left(\omega \right)=-\omega^{2}f\left(\omega \right){\left(-\omega^{2}M+i\omega C+K\right)}^{-1}.$$

In this case the calculations were carried out using a Nastran Solver (SOL108).

### 2.3. Dynamic Tests

To verify the established finite element models the experimental tests were carried out as an impulse test. The structure was excited in three perpendicular directions using a Kistler 9728A20000 modal hammer with 1.5 kg head mass (Kistler Instrumente GmbH, Sindelfingen, Germany). The responses were measured using the PCB triaxial ICP accelerometers, model 356A01 (PCB Piezotronics, Depew, New York, NY, USA). The arrangement of measuring points was depicted in Figure 3. In the case of the frame, the measurement included 233 points (marked in emerald) in the case of the frame with the robot 279 points (233 points for the frame and 46 points for the robot—marked in light green). The measurements of the responses of the analyzed structures were carried out using partial experiments involving measurements at 8 points simultaneously (measurements were carried out in this way due to the limited number of measurement channels of the frontend used).

Data acquisition was conducted using the Scadas Mobile Vibco and Testlab 2019.1 software (Siemens AG, Munich, Germany). The estimation of frequency response function was performed with the use of the H_1_ estimator. Detailed information related to the processing of the recorded signals were as follows: sampling rate 4096 Hz, frequency resolution 0.5 Hz, number of averages 10.

As a result of the impulse test conducted, 699 frequency response functions for frame and 837 for frame with a robot were determined. The parameters of the modal models were estimated using the Polymax algorithm [24,25]. The obtained modal models were validated using the modal assurance criterion (MAC), by eliminating interdependent vectors in the mode shape (the limit value of 10% was assumed) [26,27].

## 3. Results and Discussion

A comparison of the natural frequencies determined in the finite element model analysis, with the results of the experimental studies, is presented in Table 2. The agreement of the natural frequencies in is presented as the relative error δ, defined in the following way:(18)$$\delta =\left|\frac{f_{exp}-f_{FEM}}{f_{exp}}\right|\cdot 100\%,$$
where: $f_{exp}$—experimentally determined natural frequency; $f_{FEM}$—finite element model natural frequency.

The comparison of the calculated and experimentally verified mode shapes is presented in Figure 4.

When analyzing the obtained results, it can be noticed that a high compliance of the natural frequency values was obtained, in the case of the frame the maximum relative error did not exceed 10.0%, 4.3% on average; in the case of a frame with a robot, the maximum error did not exceed 19.0%, 5.6% on average. In both analyzed cases, full compliance of the mode shapes in the analyzed frequency range was achieved.

The largest discrepancies in the natural frequencies can be observed in case of modes related with the frame vibrations in the Z direction, i.e., the fifth, sixth, and seventh mode for the frame (with relative errors 7.1%, 5.8%, and 10%, respectively) and the fifth and sixth mode for the frame with the robot (with relative errors 19.0% and 10.4%, respectively). However, a more detailed analysis of dynamic properties, concerning receptance, showed that in Z directions resonance amplitudes are substantially lower than in X and Y directions. Figure 5, Figure 6 and Figure 7 show the graphs of the receptance function for the analyzed variants, where the abovementioned discrepancies in Z direction are emphasized.

When analyzing the accuracy of the mapping of the receptance function, it can be seen that the models satisfactorily reflect their real counterparts in the qualitative sense. It is related to the achieved structural compliance of the model, and thus the compliance of the mode shapes. Moreover, the model is characterized by a satisfactory ability to reproduce the damping properties of the structure. This is especially evident in the case of the amplitudes of the receptance functions for resonance frequencies. The quantitative comparison (in terms of relative differences between the amplitudes of the receptance function in resonances) shows that the best mapping was obtained for the X direction, where the relative error values were respectively 17% for the frame and 76% for the frame with the robot. In case of the Y direction, the relative error for the frame was 26%, and for the frame with robot it was 181%. The highest average error was obtained for the Z direction, the average error for the frame was 145%, and for the frame with the robot was 370%.

Nevertheless, it can be noticed that in the case of the X and Y directions the model mapping is at a high level, both in terms of the character of receptance as well as the agreement of amplitudes in resonances.

Analyzing the obtained results, it can be seen that for the frame with the attached robot, the first two mode shapes tend to vibrate in X and Y directions, respectively. These are so-called rocking vibrations low-frequency modes, which manifest themselves in significant amplitudes of the receptance functions. They are particularly important from the point of view of the proper operation of the robot. To be precise, movements performed by the robot as part of pick-and-place operations can excite these mode shapes, which in turn will lead to the movement of robot attachment, causing errors in the robot’s positioning [28].

The third mode shape is characterized by torsional vibrations of the structure around the Z axis. This mode, like the previous ones, may lead to a reduction in the accuracy of the robot’s positioning, in particular when the end-effector deviates from the robot’s vertical axis.

The remaining determined modes are mainly related to the vibrations of the steel rim in the Z direction and as in the previous analyzed cases, they may affect the accuracy of the robot’s positioning. However, they are of less importance that first three modes in terms of receptance amplitudes.

## 4. Conclusions

The paper presents the methodology for modeling the dynamic properties of a delta robot with a steel frame. The developed methodology is based on the finite element method and the Guyan reduction. The established model was used to determine the frame dynamic properties understood as values of natural frequencies, mode shapes, and frequency response functions. The two-stage verification, considering the frame itself and the frame with the mounted robot, proved the high accuracy of modeling results, when compared with the experimental test results. The maximum relative error in the case of the frame did not exceed 10.0%, 4.3% on average; in the case of a frame with a robot, the maximum error did not exceed 19.0%, 5.6% on average. In both analyzed cases, full compliance of the mode shapes in the analyzed frequency range was achieved.

Analogous comparison in case of the receptance function in resonances shows that the best mapping was obtained for the X direction, where the relative error values were respectively 17% for the frame and 76% for the frame with the robot. In the case of the Y direction, the relative error for the frame was 26%, and for the frame with a robot, it was 181%. The highest average error was obtained for the Z direction, the average error for the frame was 145%, and for the frame with the robot, 370%.

The main limitation of the proposed methodology is the poor mapping of the natural frequencies’ values related to the mode shapes characterized by main movement in the Z direction. However, due to the low significance of the Z receptance in comparison to other directions, this limitation should not be treated as eliminating the method from use. Nevertheless, it should be considered in the future work. The future work will include the use of a model updating algorithm (including a complex sensitivity analysis) to increase the accuracy of the model.

The developed methodology will serve as a tool for designing robot frames in further works. The design will focus on minimizing the robot’s vibrations by changing the frame structure. It is planning to apply a passive dampers (based, for example, on the use of a polymer concrete filling of the structure [29]) or active ones in the form of an attached external actuator system [30].

## Figures and Tables

**Figure 1 materials-15-06797-f001:**
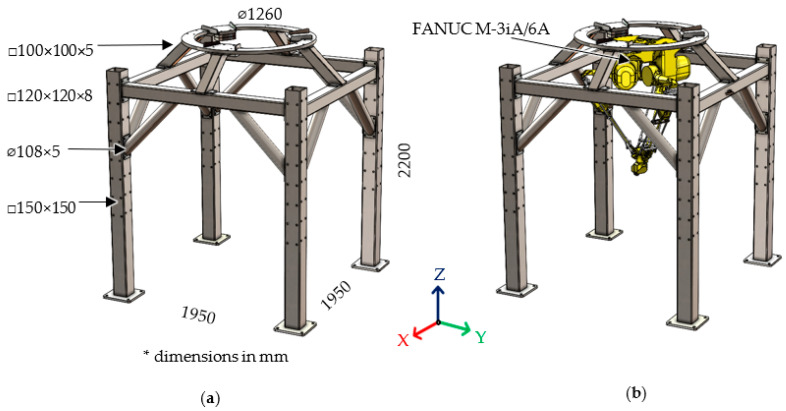
Analyzed frame (**a**) dimensions and (**b**) robot assembly.

**Figure 2 materials-15-06797-f002:**
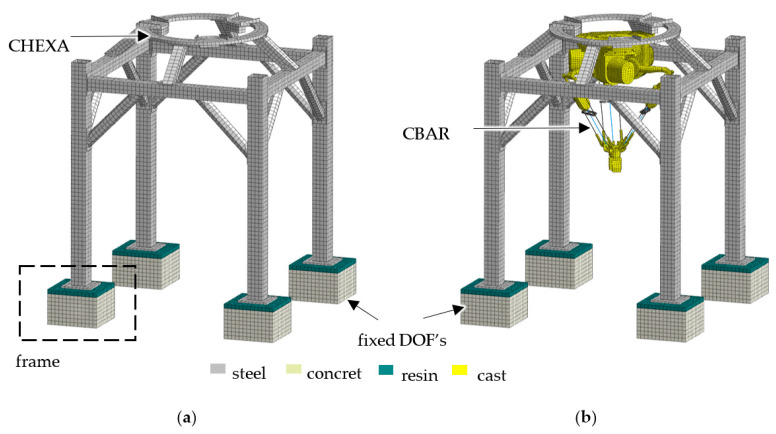
Finite element model of (**a**) the frame and (**b**) the frame with the robot.

**Figure 3 materials-15-06797-f003:**
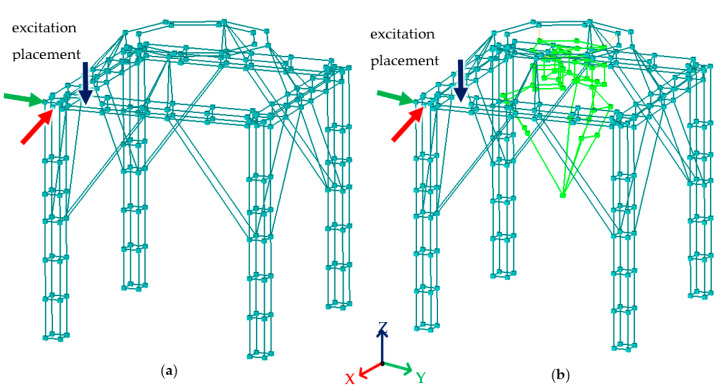
Measurement points arrangement (**a**) for the frame and (**b**) for the frame with the robot.

**Figure 4 materials-15-06797-f004:**
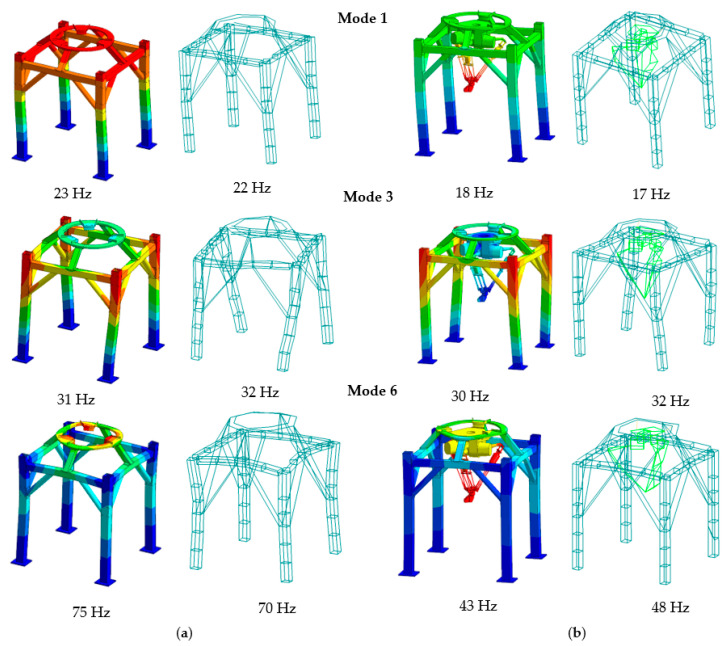
Selected mode shapes comparison (**a**) for the frame and (**b**) for the frame with the robot.

**Figure 5 materials-15-06797-f005:**
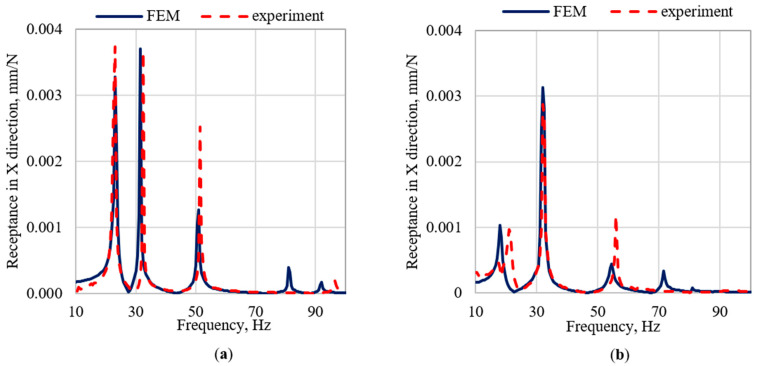
Comparison of selected receptance functions in X direction (**a**) for the frame and (**b**) for the frame with the robot.

**Figure 6 materials-15-06797-f006:**
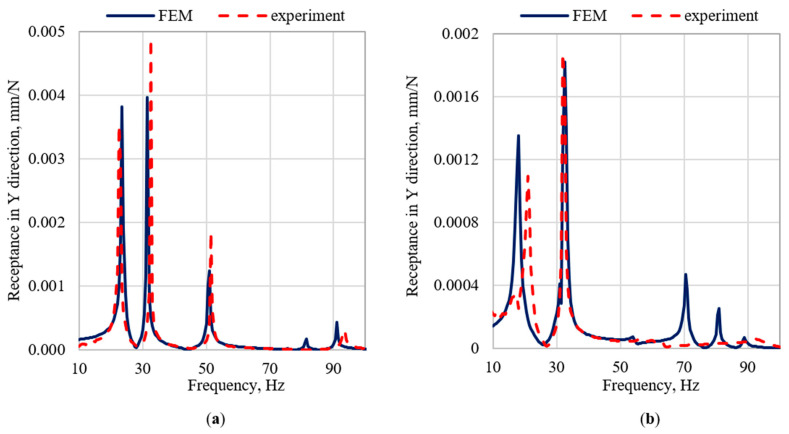
Comparison of selected receptance functions in Y direction (**a**) for the frame and (**b**) for the frame with the robot.

**Figure 7 materials-15-06797-f007:**
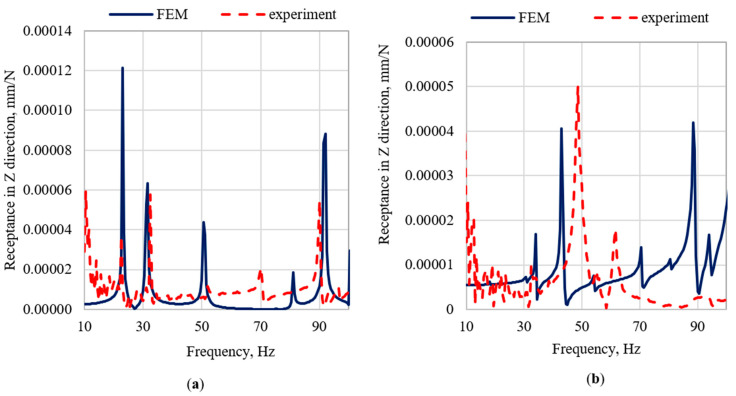
Comparison of selected receptance functions in Z direction (**a**) for the frame and (**b**) for the frame with the robot.

**Table 1 materials-15-06797-t001:** Properties of materials included in the analyzed structure.

Property	Steel	Gray Cast Iron	Concrete	Resin
Modulus of elasticity, E	210 ± 5 GPa	100 ± 3 GPa	18.3 ± 0.2 GPa	7.1 ± 0.05 GPa
Poisson’s ratio, υ	0.28 ± 0.03	0.26 ± 0.03	0.15 ± 0.02	0.3 ± 0.05
Density, ρ	7812 ± 35 kg/m^3^	6958 ± 30 kg/m^3^	2400 ± 6 kg/m^3^	2200 ± 6 kg/m^3^
Loss factor, η	0.001	0.003	0.04	0.01

**Table 2 materials-15-06797-t002:** Experimental verification of the natural frequencies of the frame and the frame with the robot.

Mode	Frame			Frame with Robot		
	Experiment	FEM	Relative Error, δ	Experiment	FEM	Relative Error, δ
1.	22 Hz	23 Hz	4.5%	17 Hz	18 Hz	5.8%
2.	23 Hz	23 Hz	0.0%	18 Hz	18 Hz	0.0%
3.	32 Hz	31 Hz	3.1%	32 Hz	30 Hz	6.3%
4.	51 Hz	50 Hz	1.9%	33 Hz	32 Hz	3.0%
5.	70 Hz	75 Hz	7.1%	42 Hz	34 Hz	19.0%
6.	86 Hz	81 Hz	5.8%	48 Hz	43 Hz	10.4%
7.	90 Hz	81 Hz	10%	55 Hz	54 Hz	1.8%
8.	93 Hz	91 Hz	2.1%	73 Hz	70 Hz	4.1%
9.	96 Hz	91 Hz	4.2%	81 Hz	81 Hz	0.0%
		**On average**	4.3%		**On average**	5.6%

## Data Availability

Data available on request due to restrictions, e.g., privacy or ethical.

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
