# Peer review of "Finite Element Modeling of Dynamic Properties of the Delta Robot with Base Frame"

_materials, 2022, doi:10.3390/ma15196797_

Round 1

Reviewer 1 Report

Manuscript Number: materials-1897034

Manuscript Finite element modeling of dynamic properties of the delta robot with base frame

The paper focuses on the dynamic properties of a delta robot frame, using the finite element modeling (Guyan reduction method) and experiment, comparing the frame response with and without an attached robot, and verifying the proposed model using experiments. the error was less than 6%. The aim is to provide an efficient model for the accurate and fast determination of base-robot-object system dynamic properties. The topic is interesting. However, the language and grammar should be carefully checked throughout the manuscript.

Reviewer comments:

Some format and writing problems

1:Page 2, (Line 96): The font size of the last paragraph of the introduction should be unified.

2:Page 5, (Line 182-183): please check for formula number mistakes. How was Eqs. (19) - (20), and Eq. (22) represent? The whole paper doesn’t have these parts. These three equations names should match Eq. (9), Eq. (11), and Eq. (12)? Please check and write carefully

The following concerns are also raised:

1: For readers to quickly catch your contribution, it would be better to highlight major difficulties and challenges, and your original achievements to overcome them, in a clearer way in the abstract and introduction.

2:  The introduction section was not well organized, especially paragraphs 2-5, (Lines 31-51) that mainly introduced the current research, like an article review without its difference to highlight its own contribution. The background and relevant references provided in the introduction are few, which cannot reflect the development of this research field well. The author should rearrange and improve the logic.

3:  The principle for determining the frame materials? For example, why was the steel ring made of 30 mm thick sheet metal? optimal value? The reason for the design range of parameter values such as Modulus of elasticity (210 ± 5 GPa:) Poisson’s ratio, (?:0.28 ± 0.03) in Table 1 should be further explained. The final input parameters compared to the test in FEM were a single value or a series values? The rationale for the choice of the particular set of parameters should be explained in more detail. Have the author numerical calculations or experiments with other sets of values? What are the sensitivities of these parameters on the results?

4:  Section 2.2. (Line 119-122): these four paragraphs should be deleted or rewritten, as it is not easy to find the connection methods, or written simply as “the finite element model can be seen in detailed in the previous paper [15]”

5:  The formula such as Eq.11-12 should be further deleted because these are the basic motion equations. I suggest removing these two equations

6:  Page 4, (Line 140): The reason for choosing Eq.1 as the damping matrix? Page 4, (Line 142-144): “To sum up, the developed frame model consisted of 80,274 degrees of freedom …… for the frame with robot 164,166 degrees of freedom”, please explain the 80,274 and 164,166 degrees of freedom?

7:  Page 6, (Line 191-192) Section 2.3: “the responses were measured in 233 (for frame) and in 279 (for 191 frame with the robot) points using PCB triaxial ICP accelerometers”, Using a total of 233 and 279 accelerometers? The principle of measuring points should be explained in detail. The test was achieved single time or series group? The principle of choosing the two responses measured points 233 (for frame) and 279 (for 191 frame with the robot) points?

8:  Page 7, (Line 211): if there are criteria for the error level? Since the maximum error was 19%, this error was quite large. The reason for the error of 19% in that case of the frame with a robot in the fifth mode (42 Hz in experiment, 34 Hz in FEM)? Perhaps experimental work which also enables varying some of the parameters to confirm the numerical results could strengthen the case for such a paper.

9:  Page 8, (Line 229): I suggest adding the responses in three directions to a figure to better present and compare the trend differences. And Fig.5 lacks necessary legends and descriptions.

10: In Section 2.1, the mass and stiffness of the frame and the mass of the robot shall be given.

11: This study proves that the experiment is insufficient and additional experiments need to be provided.

All in all, I have determined there could be other journals that may better suit your work's somewhat limited novelty with respect to the delta robot with base frame. 

Reviewer 2 Report

In this paper, a Guyan method is proposed to reduce the finite element model of a delta robot frame in order to calculate the dynamics of the robot frame quickly and accurately. Firstly, the finite element modeling of robot frame is introduced in detail, including the frame structure without attached robot and with attached robot. Then the structural frequency and frequency response function are calculated based on the reduced finite element model, and the dynamic responses of the reduced model are compared with those of the unreduced model. Before this paper is considered for publication, the following problems need to be addressed.

1. In line 166, the author uses the Guyan reduction method to reduce the degrees of freedom of the dynamic system from 6468 to 120. However, the accuracy of the reduced model needs clarification. For the accuracy of Guyan reduction method and several main model reduction methods, it is recommended to refer to some articles such as (DOI: https://doi.org/10.12989/smm.2018.5.3.417). 

2. Compared to the robot system, the frame seems unmovable. It is suggested for the authors to illustrate the necessity to consider the impaction of the frame on the dynamical responses of the robot.  

3. As well known, the mode shapes of the structure are also very important dynamic characteristics. Pls explain the similarity between the mode shapes of the reduced model and the original ones.

Reviewer 3 Report

English grammar needs to be checked. The introduction is well written, but there are no numerical comparisons or in-depth analysis on the issue. The review is good, but it needs to be improved.

"Figure 2. Finite element model of the frame under analysis (a) without robot and (b) with robot." it is necessary to improve the explanation and justification of the height of the base and the material of manufacture.

the meaning of the coefficients in the formulas must be explained everything. Specify the units of measure for the coefficients.

Conclusions  - modify information to reflect findings and comparisons across indicators. What have you done and how is it better. Show both qualitative and quantitative assessments. 

Reviewer 4 Report

This manuscript focuses on finite element modeling of dynamic properties of the delta robot with base frame, which allows accurate and fast determination of base-robot-object system dynamic properties.

My comments are as follows:

- While the abstract has aimed to provide a comprehensive overview of the main contribution, there is a need to be revised so that the general reader can grasp the main idea/topic of the draft and the main contribution.The paper lacks a detailed introduction of the experimental background and the actual situation, and lacks the relevance between the actual situation and the proposed method.

- The Authors wrote: "Based on the established model the dynamic properties (i.e., natural frequencies, mode shapes and frequency response functions) of the frame in two variants were analyzed."Why are these dynamic properties introduced for analysis? What is the process of analysis? 

-It should be given, at least intuitive (heuristic) suggestions, why the validity of the can be demonstrated? How is the characteristic of high accuracy reflected?Specifically the role of 'in case of the frame' and 'in case of the frame with robot' should be discussed. 

-In the author's paper, there are a lot of problems in the layout of graphs, and even two Figure 5. "Comparison of selected frequency response functions for a frame (a) and a frame with robot (b)." and "Comparison of selected frequency response functions for a frame (a) and a frame with robot (b) (continuation). "are quite different. Please carefully check the specification of the paper.

Round 2

Reviewer 2 Report

Considering that the authors have revised their paper according to the comments  of the reviewers, it is recommended to accept this paper.

Author Response

Thank you for recommending our manuscript for publication.

Reviewer 4 Report

Thank the authors for their efforts. The authors have adequately addressed all my concerns in the review, and did a good job to revise and improve the paper. The paper now is suitable for publication in Materials in its current form.

Author Response

(The authors gave the same response as above.)
